palaeontology/taxonomy and systematics/evolution

insects, exaggerated morphology, anti-predator–morphology, Messel, Green River, Hemiptera

**Author for correspondence:**
Sonja Wedmann
e-mail: sonja.wedmann@senckenberg.de

# Bizarre morphology in extinct Eocene bugs (Heteroptera: Pentatomidae)

Sonja Wedmann[1], Petr Kment[2], Luiz Alexandre Campos[3] and Thomas Hörnschemeyer[4]

[1]Senckenberg Forschungsstation Grube Messel, Senckenberg Forschungsinstitut und Naturmuseum Frankfurt/Main, 64409 Messel, Germany
[2]Department of Entomology, National Museum, Cirkusova 1740, 193 00 Praha 9 – Horni Pocernice, Czech Republic
[3]Department of Zoology, Federal University of Rio Grande do Sul, Avenida Bento Gonçalves 9500, prédio 43435, 91501-970, Porto Alegre, Rio Grande do Sul, Brazil
[4]Johann-Friedrich-Blumenbach-Institut für Zoologie und Anthropologie, Georg-August-Universität Göttingen, 37073 Göttingen, Germany

SW, 0000-0002-9778-4125; PK, 0000-0002-7026-5691; LAC, 0000-0001-5414-8746; TH, 0000-0002-4924-5389

Newly discovered fossil bugs (Insecta: Hemiptera: Heteroptera: Pentatomidae) from the Eocene of Messel (Germany) and Green River (North America) exhibit an exaggerated morphology including prominent spiny humeral and anterolateral angles of the pronotum and a spiny lateral abdominal margin. Especially the humeral angles are unique; they consist of expansive, rounded projections with strong spines, which is a rare trait among pentatomids. A hypothesis for the function of this extreme morphology is defence against small vertebrate predators, such as birds or reptiles. The same protuberances also produce a disruptive effect camouflaging the specimen in its environment and provide additional protection. Therefore, the extreme morphology provides primary as well as secondary anti-predator defence. The morphology of *Eospinosus peterkulkai* gen. et sp. nov. and *E. greenriverensis* sp. nov. resembles that of Triplatygini, which today occur exclusively in Madagascar, as well as that of Discocephalinae or Cyrtocorinae, which today occur in the Neotropics. Due to a lack of conclusive characters, it cannot be excluded that the fossil species may represent a case of remarkable convergence and are not related to either taxon. Phylogenetic analyses using parsimony as well as Bayesian algorithms confirmed that the new genus is a member of Pentatomidae, but could not solve its phylogenetic relationships within Pentatomidae.

# 1. Introduction

Predator–prey interactions are important driving forces of evolution, and this applies strongly for the evolution of insect biodiversity. Different groups of insects and of course of other animals have developed a wide range of various defensive strategies for protection against visually hunting predators. The strategies include, for example, crypsis, aposematism and mimicry (e.g. [1–3]).

Anti-predatory defences which minimize the likelihood of physical contact between predator and prey are called 'primary defences', while 'secondary defences' begin when subjugation or contact has begun [4, pp. 1ff, 72ff]. Examples of primary defences which avoid detection are e.g. crypsis, masquerade or disruptive camouflage. Examples of secondary defences which avoid attack after detection are e.g. morphological defences like the possession of spines, chemical defences via toxins and behavioural defences like aggressive behaviour. Also, elaborate signalling defensive strategies like aposematism and mimicry belong to these secondary defences [4, pp. 72ff].

The fossil stink bugs newly described in this paper exhibit a striking exaggerated morphology in possessing prominent spiny humeral angles and large spines on the pronotum, a spinose abdominal margin and probably elevated ridges or spines on pronotum and scutellum. We interpret these structures to have defensive functions, possibly a combination of camouflage together with morphological defence, which may affect perception and ingestion by the predator; see discussion below.

The fossils originate from two geographically very dispersed Eocene fossil sites, one is Grube Messel in Europe and the other is Green River in North America. The similar morphologies of the fossil bugs suggest they are closely related, so this peculiar and seemingly innovative morphotype was widely distributed in the Eocene. Subsequently, this morphotype went extinct at least in North America and in Europe due to unknown reasons.

# 2. Results

## 2.1. Systematic palaeontology

Heteroptera Latreille, 1810
Pentatomidae Leach, 1815

   **Genus *Eospinosus* gen. nov.** (figure 1; electronic supplementary material, figures S1–S3)
   urn:lsid:zoobank.org:act:E0C4ECC8-60F3-4102-8693-B8F69F9A9E24
   *Type species. Eospinosus peterkulkai* sp. nov.
   **Etymology**. '*Eo*' from the Eocene time epoch; '*spinosus*' in reference to the spiny morphology of the body (Latin: spinosus: thorny, spiny). The gender is masculine.
   **Differential diagnosis**. Genus separated from all other extant and fossil pentatomid genera by the following characteristics (the combination of characteristics of pronotum and connexivum is autapomorphic for the genus): pronotum with strongly produced humeral angles, humeral angles basally constricted, apically protruding and circular, with several teeth; deep incision before anterolateral angles; anterolateral angles of pronotum produced with two strong teeth; connexivum with large lobate lateral projections bearing irregular teeth.
   **Description**. Body length *ca* 6.5–8.5 mm; body roughly horseshoe-shaped due to protruding humeral angles of pronotum; body completely covered with small, more or less dense punctures.
   Head: rather large, a bit wider than long; compound eyes protruding from head by most of their width; mandibular plates wide, with distal lateral edges rectangular, with or without anteocular processes; mandibular plates converging slightly towards tip of clypeus, but not meeting; clypeus rather straight, about as long as mandibular plates.
   Pronotum: most notable are the strongly protruding, basally constricted humeral angles, they have a rounded shape and show large to smaller teeth; teeth form and number vary; strong incision before anterolateral angles; anterolateral angles of pronotum are produced with two strong teeth; anterior margin of pronotum deeply arcuately concave to receive head, lateral margin slightly convex rounded, posterior margin of pronotum slightly convex rounded; perhaps with three-dimensional (3D) ridges or spines present on pronotum.
   Scutellum: large and broadly triangular, with indentations at middle of its lateral sides; apex of scutellum broadly rounded, reaching end of tergite 7; anterior half of scutellum bulged out widely, with an additional bulge or perhaps spine in the middle.

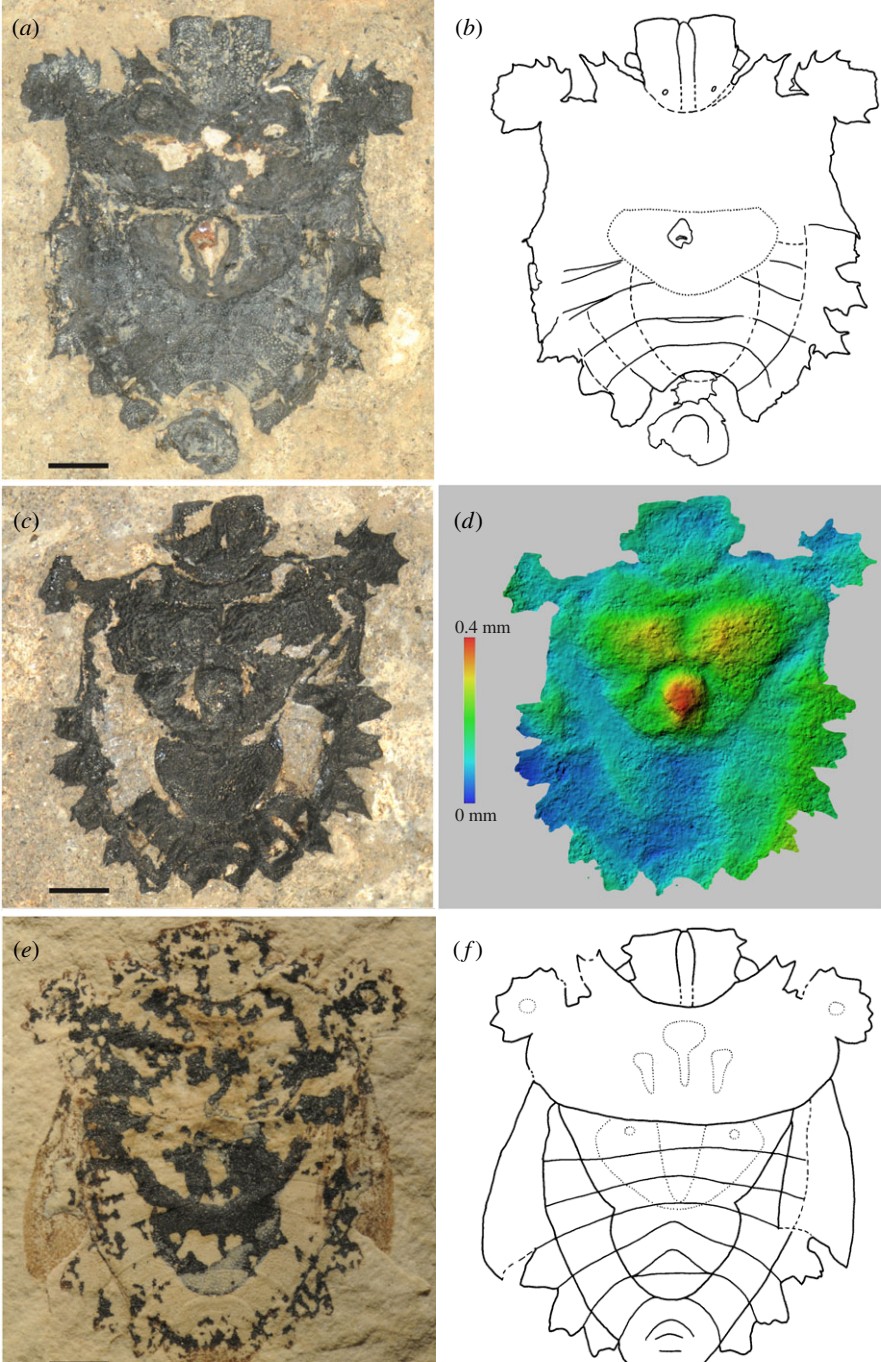

**Figure 1.** *Eospinosus peterkulkai* sp. nov. and *Eospinosus greenriverensis* sp. nov. (*a,b*) *Eospinosus peterkulkai* sp. nov., holotype, male, SF-Mel 1654. Photo and interpretative drawing of the specimen. Main structures drawn with continuous lines, incomplete or internal structures drawn with dashed lines; 3D structures dotted, (*c,d*), *Eospinosus peterkulkai* sp. nov., female, SF-Mel 15483. Photo and colour-coded 3D relief, reconstructed from photogrammetry. Elevated areas orange-red, lower areas green-blue. (*e,f*), *Eospinosus greenriverensis* sp. nov., male, holotype UCM 79057. Photo and interpretative drawing of the specimen. Main structures drawn with continuous lines, incomplete or internal structures drawn with dashed lines; 3D structures dotted. All scales 1 mm.

Thoracic venter: mesosternal carina (electronic supplementary material, figure S3: mec), ostiole of metathoracic scent glands (electronic supplementary material, figure S3: ost) and metathoracic spiracle (electronic supplementary material, figure S3: mts) well developed.

Abdomen: connexivum fully exposed dorsally, with large lobate projections bearing irregular teeth. Ventrites VI and VII medially angulated produced anteriorly, ventrite VII appearing triangular, more triangular in male than in female.

Antennae and legs not preserved; hemelytra too poorly preserved to see details.

*Eospinosus peterkulkai* **sp. nov.** (Figure 1*a–d*; electronic supplementary material: figures S1, S2*a–e*, S3 and tables S1, S2)

urn:lsid:zoobank.org:act:A8DCB0AE-A89F-4364-86C2-98F9CEF574D0

**Holotype**. SF-MeI 1654 (male, figure 1*a,b*), deposited in the Messel insect collection of the Senckenberg Research Institute Frankfurt (SF), at the Senckenberg Research Station Grube Messel.

**Etymology**. Named in honour of Peter Kulka (*1937), who was the architect responsible for the recent rebuilding of the Senckenberg Research Institute and Natural History Museum Frankfurt/M.; name in genitive case.

**Paratypes**. SF-MeI 5790 (male), SF-MeI 15483 (female), SF-MeI 16148 (male), SF-MeI 16861 (female).

**Type locality and horizon**. Messel (latitude 49°55′03.4″ N, longitude 8°45′30.6″ E) near Darmstadt, Hesse, Germany; Messel Formation, lower Mid-Eocene, Lutetian, *ca* 47.5 Ma. The holotype SF-MeI 1654 was collected in the year 1986 in the Messel pit in grid square H12, in strata 0–1 m above local stratigraphic marker level M.

**Description** (based on five fossils). As for the genus, with the following additions and specifications: body length 6.7–8.2 mm, width incl. humeral angles 6.5–6.9 mm, outer lateral margins of mandibular plates straight, without spine; incision between pronotal angle and humeral angle straight, without a spine; coloration dark, intact cuticula metallic black with a hue of blue, one specimen with a stronger hint of bluish structural coloration (SF-MeI 16861). Female terminalia, see electronic supplementary material, figure S2*a–c*. Genital capsule of male with posterolateral projections rounded not prominent; its ventral wall with semicircular impression medially (electronic supplementary material, figure S2*d–f*). Tergite VIII of female produced posteriorly in two sharp lobes. A similar morphology is shared by both female and male specimens, no sexual dimorphism can be detected.

Detailed measurements of the five fossils belonging to this species are given in electronic supplementary material, table S1.

**Differential diagnosis**: see description of *E. greenriverensis* below.

**Preservation**. All five specimens are strongly compressed rock fossils that show remarkably detailed preservation. The holotype SF-MeI 1654 (figure 1*a,b*) is rather completely preserved from the dorsal side; it shows the characters of the genus and species. It is a male specimen; the genital capsule is well recognizable. Two ocelli are visible at the base of its head. In the middle of its body, intestinal contents are preserved, these are reddish coloured, have a hard, brittle consistency. These remains are interpreted as plant sap which fossilized and became amber. The 3D structure is only moderately visible. In specimen SF-MeI 5790 (electronic supplementary material, figure S1*a*) the head is missing completely, otherwise, it is relatively completely preserved and visible from its ventral side. Its sex is male, with ventral wall of genital capsule well preserved. The high intra-individual variability of the spination of the bizarre humeral angles can be seen in this specimen when the left and right humeral angles are compared. In the meso- and metapleuron, the ostiole and metathoracic spiracle are visible (electronic supplementary material, figure S3); the abdominal spiracles, peritreme and trichobothria are indistinct. In the middle of the body, remains of intestinal contents (amber) are present. The 3D structure is hardly preserved. Specimen SF-MeI 15483 (figure 1*c,d*) is preserved from its dorsal side, the terminal appendages are quite well visible and indicate it is a female. The two strong teeth of the anterolateral angles of the pronotum are missing due to incomplete preservation. The 3D structure of pronotum and scutellum in this specimen is well preserved (figure 1*d*). Specimen SF-MeI 16148 (electronic supplementary material, figure S1*b*) is preserved from its dorsal side, the protruding terminal appendages indicate it is a male. The 3D structure of the pronotum and scutellum in this specimen is rather well preserved. SF-MeI 16861 (electronic supplementary material, figure S1*c*) is visible from its ventral side. Its sex is female, terminal appendages are recognizable, at least valvifers VIII and laterotergites IX (electronic supplementary material, figure S2*b,c*). In the ventral view of the head the mandibular plates and the maxillary plates can be discerned. In this specimen the cuticula is relatively well preserved and has a pronounced bluish tinge.

**Palaeobiology.** SF-MeI 1654 and SF-MeI 5790 contain pieces of fossilized plant sap in their gut. A small piece can be seen in figure 1*a*, the brown structure in the middle of the scutellum. This indicates that this species fed on plant sap, which during the fossilization transformed into amber.

*Eospinosus greenriverensis* **sp. nov.** (Figure 1*e,f*; electronic supplementary material, figure S2*f* and table S1)

urn:lsid:zoobank.org:act:2AC26AD9-CEB7-415D-8A7C-298DD39F5C23

**Holotype**. UCM 79057a, b (male, part and counterpart), collected on 10 July 2007 by David Kohls, deposited in UCM.

**Etymology**. From the fossil site Green River where the specimen was collected, adjective.

**Type locality and horizon**. Green River Formation, UCM locality 2005025 (Anvil points Kohls), Piceance Basin, Garfield County, Colorado, USA, Parachute Creek Member of the Green River Formation; Early Eocene (Late Ypresian).

**Description** (based on one fossil). As for the genus, with the following additions and specifications: body length *ca* 7.8 mm, width incl. humeral angles *ca* 6.7 mm, outer lateral margins of mandibular plates with a prominent anteocular process; incision between pronotal angle and humeral angle not straight, but bearing also a small process; coloration dark. Genital capsule with posterolateral projections narrowly rounded, produced posteriorly, ventral wall between them concave.

Detailed measurements are given in electronic supplementary material, table S1.

**Differential diagnosis** between *E. peterkulkai* and *E. greenriverensis*. The two species are rather similar, except for the presence of an anteocular spine on the mandibular plate (absent in *E. peterkulkai*, present in *E. greenriverensis*), structure of the border between anterolateral angle and humeral angle and shape of the genital capsule. The compound eyes of *E. greenriverensis* may be a bit larger than those of *E. peterkulkai*. Another feature might be the blunder teeth at the humeral angles in *E. greenriverensis*, but that might be a preservation artefact. The head on the first view looks to be different in length, but this is probably due to preservation.

**Preservation**. Specimen UCM 79057 a, b (figure 1*e,f*) is a strongly compressed rock fossil; it is split into two parts a and b; features from both ventral and dorsal sides are visible. The specimen shows all characters of the genus and species. The 3D structure of pronotum and scutellum in this specimen is rather well preserved. Original coloration is not preserved, remains of the fossil cuticle on the anterior portion of the hemelytra have a light brown colour, the rest of the body has a dark brown colour.

## 2.2. Phylogenetic relationships

The placement of *Eospinosus* gen. nov. within Pentatomidae appears unequivocal and is corroborated by the phylogenetic analyses (figure 2). However, due to the preservation of the fossils, several character states cannot be determined. This leads to a very large number of equally parsimonious trees in the parsimony analysis (432.156 trees of 246 steps each, found in 69 islands) and several polytomies in the consensus tree (figure 2*a*). For Pentatomidae only the monophyly of the group is confirmed but within the family mainly a large polytomy remains. The Bayesian analysis produces a very similar result (two parallel runs with a standard deviation of split frequencies of 0.003643, PSRF = 1.0 after 10 million generations, figure 2*b*) with a large polytomy within Pentatomidae.

Besides *Eospinosus* gen. nov., also *Acanthocephalonotum martinsnetoi* Petrulevičius and Popov, 2014 from the Middle Eocene of Patagonia, Argentina, is consistently placed in the polytomy within Pentatomidae. Therefore, the placement of *A. martinsnetoi* within Discocephalinae, as proposed by Petrulevičius & Popov [5] cannot be confirmed.

The characteristic shape of the head and the presence of the spinose humeral angles of the pronotum and connexivum are similar to other representatives of Pentatomidae, especially Pentatominae: Triplatygini or Cyrtocorinae or Discocephalinae.

Due to the results of the phylogenetic analyses and the difficulties of identifying unequivocal characters associating the fossil species with one of the subgroups of Pentatomidae, we refrain from assigning *Eospinosus* gen. nov. to any of the subfamilies. Anyway, recent DNA-based phylogenetic analyses suggest the current system of subfamilies and tribes of Pentatomidae as unsatisfactory and we may expect dramatic changes in its classification in the near future (e.g. [6–10]).

# 3. Discussion

## 3.1. Evolution of bizarre morphology

Which kind of selection shaped the evolution of the elaborate spiny humeral angles of the pronotum and the spiny abdomen of these bugs, and, accordingly, what function did they have? Morphological structures such as sharp spines are common in very different phyla of animals, and it seems very likely that these structures function as anti-predator defences [4].

Our assumption is that due to their shape and structure the bizarre protuberances in the fossil bugs have evolved to serve anti-predatory defence functions.

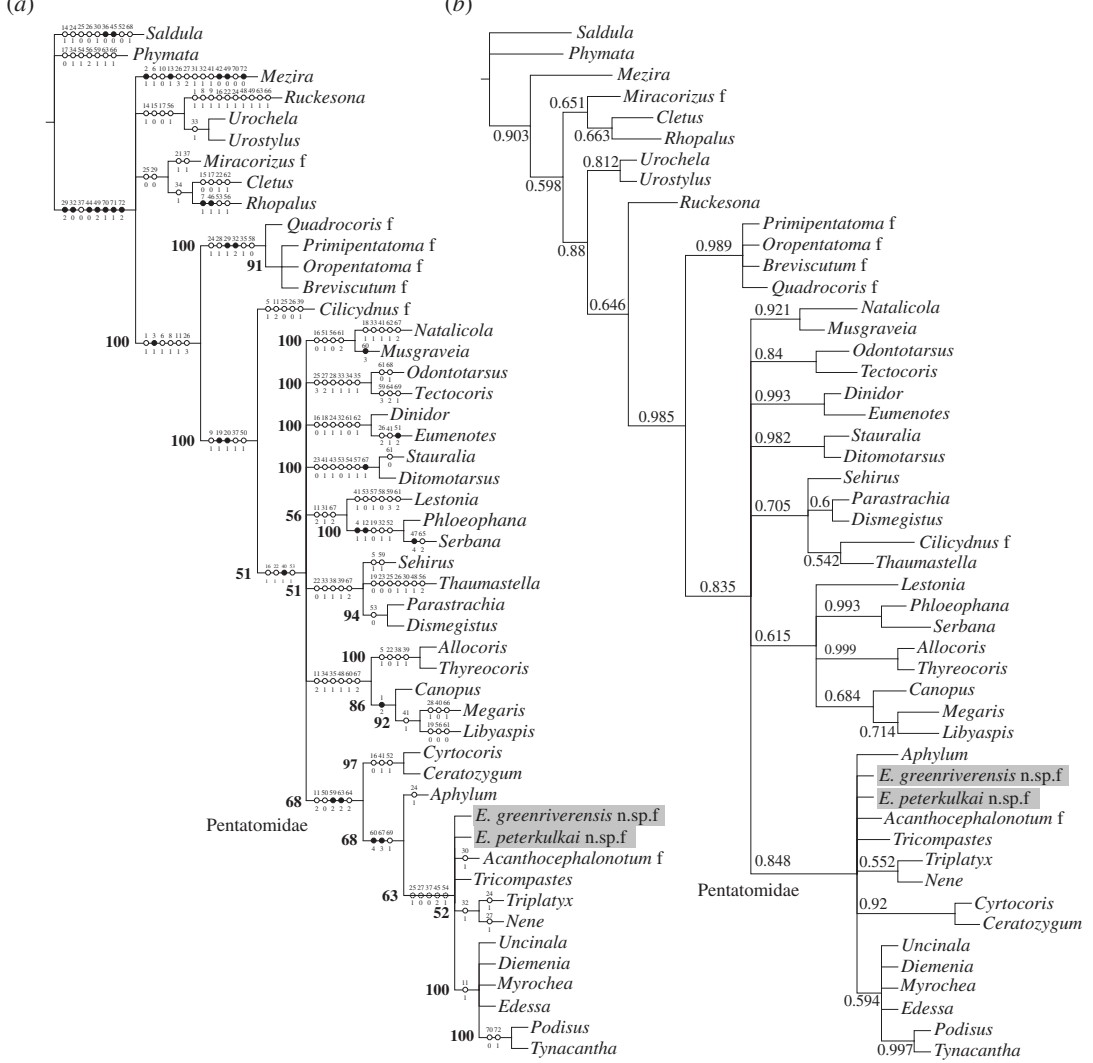

**Figure 2.** Results of phylogenetic analyses. (*a*) 50% majority rule consensus tree of 432 156 equally parsimonious trees with 246 steps each from PAUP\* (hs, tbr, random addition, 1000 replications) with potential autapomorphies (character number above dot, character state below dot). Numbers in bold = percentage of occurrence of node in resulting trees. (*b*) consensus tree from MrBayes (10 million generations) with posterior probabilities.

Of course, not only defence drives the evolution of bizarre shapes in insects, or generally in animals. Another hypothesis might be that these structures could have had their origin via sexual selection. What contradicts that possibility is that exaggerated traits that function as signals in the context of sexual selection are usually very variable in their expression ([11], figure 2]), which is not the case in the fossils. Besides, the presence of sexual dimorphism would hint to that kind of driving force (e.g. [12,13]). For *Eospinosus* gen. nov. we can exclude sexual dimorphism, because we found female and male specimens with the same morphology in the species *E. peterkulkai* sp. nov. So all in all, the defence hypothesis is supported.

Projections and processes break the characteristic outline of the body, and so can conceal the body contour from visually hunting predators [2, p. 15]. We think that the same applies to the fossil bugs: the exaggerated protuberances produce a disruption of the body outline, thereby camouflaging the specimen in its environment.

Concerning a second line of defence, for example, Young *et al.* [14] quantified the selection of a bird predator on relative horn lengths of a lizard species. They demonstrated convincingly that defence against bird predation drove the elongation of horns in the investigated lizards. In certain katydids, especially the so-called armoured ground 'crickets' or the spiny devil katydids of the genus *Panacanthus* Walker, 1869, large spines on the pronotum are interpreted to make them 'a very disagreeable food item for many vertebrates' and handling becomes more difficult, too [15, p. 83]. The

function of spines in the spiny devil katydids is interpreted as anti-predator defence, possibly mainly against daytime predators, but direct evidence for this is still missing [16, p. 51]. We also interpret the spines and protuberances in the fossil species to complicate predator handling and ingestion of the bugs, and in combination with secretion of their repugnatorial glands (their function is confirmed by the presence of an ostiole), they could have been an effective defence mechanism. Possible predators might have been small vertebrates such as birds or reptiles.

So, in conclusion, we think that the exaggerated spiny structures serve both primary defence by providing camouflage and secondary defence by physically hindering the process of being eaten.

More or less similar exaggerated morphologies occur in several different groups of Pentatomoidea, for example in some pentatomoid families (e.g. Scutelleridae, Tessaratomidae, Urostylidae) and also in several subgroups of Pentatomidae (e.g. in Cyrtocorinae, Triplatygini and Discocephalinae). An interesting question is, where and when did these structures evolve? One assumption is that these structures evolved independently in different subgroups of Pentatomidae. Another, in our opinion less likely interpretation, could be that this kind of morphology evolved already in the last common ancestor of Pentatomidae with subsequent modifications. To be able to decide this question, a comprehensive phylogenetic analysis focusing on this kind of traits among pentatomoids would be necessary, which is beyond the scope of this paper.

## 3.2. Palaeobiogeography and lifetime of species

Extant pentatomid bugs showing exaggerated spinose morphologies are found especially in Cyrtocorinae, Discocephalinae and Triplatygini. Triplatygini is endemic to Madagascar [17–21], while Cyrtocorinae and Discocephalinae are endemic to the subtropical and tropical regions of the New World [22]. This biogeographic distribution is intriguing, because during the Eocene, the fossil genus *Eospinosus*, which also has a bizarre spinose morphology, was apparently widely distributed in the Northern Hemisphere. The climate in the Northern Hemisphere during the Eocene was warm and equable (e.g. [23]), and there is evidence from other fossil insect groups to have had a former Holarctic distribution during the Eocene (e.g. [24]). Today, this morphotype is not present in the extant bug fauna of the Holarctic region, it went extinct sometime after the Eocene.

A minimum lifespan of 171 000 years for the species *Eospinosus peterkulkai* sp. nov. from Messel can be calculated. Specimens of this species were found around local marker horizon M and at a minimum of 0.52 m below local marker horizon alpha (see electronic supplementary material, table S2). Since the distance between the marker horizons M and alpha is 23.46 m [25], this results in a distance of at least 23.98 m between the oldest and the youngest fossils of *E. peterkulkai* sp. nov. With an average sedimentation rate of 0.14 mm per year [26], this equals a time span of around 171 000 years.

# 4. Material and methods

The described specimens originate from the Messel Formation, Grube Messel, near Darmstadt, Hesse, Germany, and from the Green River Formation, Garfield County, Colorado, USA. They are deposited in the Senckenberg Forschungsstation Grube Messel of the Senckenberg Forschungsinstitut und Naturmuseum Frankfurt, Germany (SF) and in the Museum of Natural History of the University of Colorado, Boulder, USA (UCM).

The fossils were studied under binocular stereomicroscopes Leica MZ125 and Leica M165C, drawn with the aid of a drawing attachment and photographed using a Nikon D300 attached to Leica MZ125. Helicon Focus v. 5.3 was used to stack images.

Measurements in electronic supplementary material, table S1 were made with an ocular micrometer attached to MZ125.

Measurements were taken as follows: body length: along midline from apex of clypeus to tip of abdomen in females, to tip of pygophore/tip of t7 in males; head length: along midline from clypeus to the anterior pronotal margin; head width: maximum width anterior of eyes; pronotum length: medially; pronotum width including humeral angles: maximum width; pronotum width below humeral angles: width posterior of crownlike humeral angles; scutellum length: medially from base to apex; scutellum width: maximum width at base; abdomen width: maximum width as preserved, with and without spines.

General morphological terminology follows Tsai *et al.* [27].

## 4.1. Photogrammetry

The fossils show a relief in the body, especially fossil SF-MeI 15483. To visualize and analyse this relief photogrammetry was applied. For photogrammetric reconstruction, a set of 38 images were taken. Twenty-eight images were taken while rotating the specimen in front of the camera. The remaining 10 images were taken while moving the specimen longitudinally and transversally in front of the camera with each image capturing greater than or equal to 50% of the fossil. An Olympus Tough TG-5 set to 'Microscope' mode was used for taking the images with a slightly oblique view on the fossil. Reconstruction of a 3D model and the colour-coded image shown in figure 1*d* was done with Agisoft Metashape Professional v. 1.6.3.

## 4.2. Fossil sites

Grube Messel near Frankfurt am Main, Hesse, Germany, is an internationally renowned Konservat-Lagerstätte recognized by UNESCO as a world heritage site. The preservation of its fossils is extraordinary and gives detailed insights into a Middle Eocene (Lutetian) paratropical arboreal environment (e.g. [28]). Best known are the vertebrates, but ecologically just as important is the insect fauna. A wide range of different groups of insects has been documented from Messel [29], including e.g. the first fossil leaf insect and a very diverse hymenopteran fauna [30–33]. Permission for digging for fossils in the Messel Pit Fossil Site was issued to the Senckenberg Research Institute Frankfurt by the following authority: Archaeologische und palaeontologische Denkmalpflege, hessenArchaeologie, Schloss Biebrich/Ostfluegel, 65203 Wiesbaden, Germany.

The Green River Formation, located in western North America, comprises extensive lacustrine deposits in which many exceptionally preserved fossils have been found. A diverse biota has been recovered from the Green River formation (e.g. [34,35]). Insects from the northern deposits of the Green River Basin have been studied since the late nineteenth century [36]. Insects from the southern Piceance Basin of former Lake Uinta have been studied only more recently (e.g. [37–41]). The fossil studied in this paper was found in sediments from the southern Piceance Basin which belong to the latest Early Eocene [42,43]. No special permits were required for collecting this insect fossil.

## 4.3. Phylogenetic analysis

To determine the phylogenetic position of the fossils we used the dataset published by Yao *et al.* [44]. Their matrix comprises 72 morphological characters and 40 species from six outgroup taxa: *Saldula brevicornis* Rimes, 1951 (Saldidae), *Phymata pennsylvanica* Handlirsch, 1897 (Reduviidae), *Mezira sayi* Kormilev, 1982 (Aradidae), *Cletus punctiger* (Dallas, 1852) (Coreidae), *Miracorizus punctatus* Yao, Cai and Ren, 2006 (Middle Jurassic fossil Rhopalidae) and *Rhopalus latus* (Jakovlev, 1883) (Rhopalidae), and 34 ingroup taxa comprising five fossil and 29 extant species from 19 families. We supplemented this matrix with five further pentatomid species: *Ceratozygum horridum* (Germar, 1839) (Cyrtocorinae), *Edessa rufomarginata* (De Geer, 1773) (Edessinae), *Triplatyx stysi* Kment, 2008, *Tricompastes gigas* Cachan, 1952, *Nene undulatum* Kment, 2015 (all three Pentatominae: Triplatygini) and the fossil *Acanthocephalonotum martinsnetoi* Petrulevičius and Popov, 2014. The final data matrix comprised 49 species and 72 characters (see electronic supplementary material, table S3: list of taxa; electronic supplementary material, table S4: character descriptions; data matrix for phylogenetic analysis).

For a few species and characters, we corrected the coding where we noted inconsistencies in the original matrix of [44].

The matrix was managed and edited with Mesquite 3.61 [45]. The reconstruction of the phylogenetic relationships was done with a parsimony approach using PAUP* 4a168 [46] as well as with a Bayesian analysis using MrBayes 3.2.7a [47,48]. The resulting trees were edited using ASADO v. 1.61 [49] and Inkscape 1.0.2 (GNU GPLv2, https://inkscape.org/).

In the parsimony analysis, all characters were set to unordered (=non-additive). Six species were defined as outgroup as explained above. Due to the size of the matrix, heuristic search was used for tree reconstruction with parameters set as follows: swap = TBR, addSeq = random, nreps = 1000. Further, 'SET maxtrees = 200, increase = auto, autoinc = 100'; was used to guarantee enough storage even for large numbers of trees during tree search. With these settings, taxa are added in randomized sequence during the start of the tree search. Occasionally, this can lead to extremely long search times. To achieve results in reasonable time we limited the maximum number of rearrangements in the TBR (abortrep = yes, rearrlimit = 100 000 000, limitperrep = yes). In the final analysis, this limit was hit in

217 out of 1000 replicates with a total of $2.6661 \times 10^{10}$ tried rearrangements. Due to the random addition, these values can vary strongly with each new analysis. The remaining parameters were left with the defaults.

In the Bayesian analysis, the standard model for morphological characters as implemented in MrBayes and proposed by Lewis [50] and Nylander *et al.* [51] was employed in its simplest version, with all state frequencies (change rates) set equal, all topologies with equal probabilities, and with unconstrained branch length. Since MrBayes allows for one outgroup taxon only, *Saluda brevicornis* was set as single outgroup. This taxon was chosen because according to Weirauch *et al.* [52] it represents the most basal heteropteran taxon available in our matrix. The analysis was run over 10 million generations.

## 4.4. Nomenclatural acts

The nomenclatural acts in this work have been registered in Zoobank. The relevant LSIDs can be found in the sections describing the taxa.

Data accessibility. All data concerning additional morphological information and all data concerning the phylogenetic analysis are available as electronic supplementary material [53].

Authors' contributions. S.W. and P.K. designed research; S.W., P.K., L.A.C. and T.H. performed research, analysed data and wrote the paper.

Competing interests. We declare we have no competing interests.

Funding. The present study was financially supported by grants of the Deutsche Forschungsgemeinschaft (DFG, grant nos. WE 2942/6-1 and WE 2942/6-2 to S.W.). The work of P.K. was financially supported by the Ministry of Culture, Czech Republic (grant nos. DKRVO 2019–2023/5.I.c, MK000023272, to National Museum, Praha). Grant support was given to L.A.C. by the Conselho Nacional de Desenvolvimento Científico e Tecnológico (CNPq), Brasil (grant no. 310933/2018-8).

Acknowledgements. We sincerely thank David Kohls, Battlement Mesa, Colorado, USA, for collecting the insects from Green River. Many thanks to Dena Smith (now National Science Foundation) and Talia Karim (UCM) for the loan of the fossil bug from Green River. We thank all members of the Senckenberg digging teams for their efforts and Uta Kiel (SF) for the careful preparation of the fossil Messel insects. We thank Christiane Weirauch, University of California Riverside, USA, for advice on an early draft of the manuscript, and we thank Julian F. Petrulevičius, Universidad Nacional de La Plata, Argentina and the anonymous reviewers for their advice and comments, which helped to improve the manuscript. The publication of this article was funded by the Open Access Fund of the Leibniz Association.

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
