## [Peer Review File · Royal Society Open Science]

Review History

RSOS-211466.R0 (Original submission)

Review form: Reviewer 1

Is the manuscript scientifically sound in its present form?

Yes

Are the interpretations and conclusions justified by the results?

Yes

Is the language acceptable?

Yes

Do you have any ethical concerns with this paper?

No

Have you any concerns about statistical analyses in this paper?

No

Recommendation?

Accept with minor revision (please list in comments)

Comments to the Author(s)

Please check the use of e.g., and e.g. (sometimes with comma sometimes without)

Photogrammetry: Why was it used? Not really clear to me? Any special application for the description of the fossil specimen. I could not find anything related to this in the species description?

p.21, line 434: citation for Grande 1984 has to be placed in the next line

p. 16, line 290: The distance between the marker horizons divided by the sedimentation rate is not equal to 171,000 years but only 167,571 years

Review form: Reviewer 2

Is the manuscript scientifically sound in its present form?

Yes

Are the interpretations and conclusions justified by the results?

Yes

Is the language acceptable?

Yes

Do you have any ethical concerns with this paper?

No

Have you any concerns about statistical analyses in this paper?

No

Recommendation?

Accept as is

Comments to the Author(s)

Congratulations to the authors, nice paper in continuation of the research on the southern *Acanthocephalonotum* describing this northern genus. Also hypothesising the phylogenetic relationships of these suggestively similar genera! I hope we could continue dialoguing through new research in the future for better knowledge of these Eocene pentatomids! all the best, Julian

Decision letter (RSOS-211466.R0)

Dear Dr Wedmann

On behalf of the Editors, we are pleased to inform you that your Manuscript RSOS-211466 "Bizarre morphology in extinct Eocene bugs (Heteroptera: Pentatomidae)" has been accepted for publication in Royal Society Open Science subject to minor revision in accordance with the referees' reports. Please find the referees' comments along with any feedback from the Editors below my signature.

Please submit your revised manuscript and required files (see below) no later than 7 days from today's (ie 28-Oct-2021) date. Note: the ScholarOne system will 'lock' if submission of the revision is attempted 7 or more days after the deadline. If you do not think you will be able to meet this deadline please contact the editorial office immediately.

on behalf of Kevin Padian (Subject Editor)
openscience@royalsociety.org

Associate Editor Comments to Author:

Comments to the Author:

A couple of relatively minor matters have been identified as worth addressing by one of the reviewers, but overall the view is very positive - well done and we'll look forward to receiving the revision soon!

Reviewer comments to Author:

Reviewer: 1

Comments to the Author(s)

please check the use of e.g., and e.g. (sometimes with comma sometimes without)

Photogrammetry: Why was it used? Not really clear to me? Any special application for the description of the fossil specimen. I could not find anything related to this in the species description?

p.21, line 434: citation for Grande 1984 has to be places in the next line

p. 16, line 290: The distance between the marker horizons divided by the sedimentation rate is not equal to 171,000 years but only 167,571 years

Reviewer: 2

Comments to the Author(s)

Congratulations to the authors, nice paper in continuation of the research on the southern *Acanthocephalonotum* describing this northern genus. Also hypothesising the phylogenetic relationships of these suggestively similar genera! I hope we could continue dialoguing through new research in the future for better knowledge of these Eocene pentatomids! all the best, Julian

===PREPARING YOUR MANUSCRIPT===

one version should clearly identify all the changes that have been made (for instance, in coloured highlight, in bold text, or tracked changes);

===PREPARING YOUR REVISION IN SCHOLARONE===

-- If you are requesting an article processing charge waiver, you must select the relevant waiver option (if requesting a discretionary waiver, the form should have been uploaded, see 'File upload' above).

-- If you have uploaded any electronic supplementary (ESM) files, please ensure you follow the guidance at <https://royalsociety.org/journals/authors/author-guidelines/#supplementary-material> to include a suitable title and informative caption. An example of appropriate titling and captioning may be found at https://figshare.com/articles/Table_S2_from_Is_there_a_trade-off_between_peak_performance_and_performance_breadth_across_temperatures_for_aerobic_scope_in_teleost_fishes_/3843624.

Author's Response to Decision Letter for (RSOS-211466.R0)

See Appendix A.

Decision letter (RSOS-211466.R1)

Dear Dr Wedmann,

I am pleased to inform you that your manuscript entitled "Bizarre morphology in extinct Eocene bugs (Heteroptera: Pentatomidae)" is now accepted for publication in Royal Society Open Science.

Please also send an updated email address for your colleague with the account 'sigara@post.cz' as this is not currently able to receive messages from the journal, which is a requirement for publication.

on behalf of Prof Kevin Padian (Subject Editor)
openscience@royalsociety.org

Appendix A

Reply to reviewers' comments:

Reviewer: 1

Comments to the Author(s) please check the use of e.g., and e.g. (sometimes with comma sometimes without) – **ok, done**

Photogrammetry: Why was it used? Not really clear to me? Any special application for the description of the fossil specimen. I could not find anything related to this in the species description?) – **As mentioned under „4. Material and Methods“, „Photogrammetry“, the method was applied to visualize the relief of the specimen. In a standard 2D photograph the relief is not clearly visible. The color-coded reconstruction from the photogrammetry (fig. 1d), however, gives a good impression of most of the 3D features.**

p.21, line 434: citation for Grande 1984 has to be placed in the next line – **ok, done**

p. 16, line 290: The distance between the marker horizons divided by the sedimentation rate is not equal to 171,000 years but only 167,571 years – **calculation of the vertical distance between fossils is explained now in more detail, so it becomes clear why it is around 171,000 years.**

Reviewer: 2

Comments to the Author(s) Congratulations to the authors, nice paper in continuation of the research on the southern Acanthocephalonotum describing this northern genus. Also hypothesising the phylogenetic relationships of these suggestively similar genera! I hope we could continue dialoguing through new research in the future for better knowledge of these Eocene pentatomids! all the best, Julian – **yes, great, many thanks!**